# Brown Seaweed Food Supplementation: Effects on Allergy and Inflammation and Its Consequences

**DOI:** 10.3390/nu13082613

**Published:** 2021-07-29

**Authors:** Simone E. M. Olsthoorn, Xi Wang, Berend Tillema, Tim Vanmierlo, Stefan Kraan, Pieter J. M. Leenen, Monique T. Mulder

**Affiliations:** 1Laboratory of Vascular Medicine, Department of Internal Medicine, Erasmus MC, University Medical Center, 3015 GD Rotterdam, The Netherlands; olsthoorn98@gmail.com (S.E.M.O.); wangxihnzyu@gmail.com (X.W.); 2Department of Immunology, Erasmus MC, University Medical Center, 3015 GD Rotterdam, The Netherlands; p.leenen@erasmusmc.nl; 3The Basic Medical Laboratory, Hunan University of Chinese Medicine, Changsha 410208, China; 4Seaweed Food Solutions BV, 8444 DC Heerenveen, The Netherlands; tillema@boxnv.nl; 5Neuroimmune Connections & Repair (NIC&R), Biomedical Research Institute (Biomed), Hasselt University, B-3590 Diepenbeek, Belgium; tim.vanmierlo@uhasselt.be; 6Department of Immunology and Biochemistry, Biomedical Research Institute (Biomed), Hasselt University, B-3590 Diepenbeek, Belgium; 7R&D Unit, Carnmore, Oranmore, H91 E09X Galway, Ireland; stefan.kraan@theseaweedcompany.com; 8The Seaweed Company Blue Turtle, Co., Ltd., H91 E09X Galway, Ireland

**Keywords:** seaweed, allergy, inflammation, oral

## Abstract

Multiple health benefits have been ascribed to brown seaweeds that are used traditionally as dietary component mostly in Asia. This systematic review summarizes information on the impact of brown seaweeds or components on inflammation, and inflammation-related pathologies, such as allergies, diabetes mellitus and obesity. We focus on oral supplementation thus intending the use of brown seaweeds as food additives. Despite the great diversity of experimental systems in which distinct species and compounds were tested for their effects on inflammation and immunity, a remarkably homogeneous picture arises. The predominant effects of consumption of brown seaweeds or compounds can be classified into three categories: (1) inhibition of reactive oxygen species, known to be important drivers of inflammation; (2) regulation, i.e., in most cases inhibition of proinflammatory NF-κB signaling; (3) modulation of adaptive immune responses, in particular by interfering with T-helper cell polarization. Over the last decades, several inflammation-related diseases have increased substantially. These include allergies and autoimmune diseases as well as morbidities associated with lifestyle and aging. In this light, further development of brown seaweeds and seaweed compounds as functional foods and nutriceuticals might contribute to combat these challenges.

## 1. Introduction

Brown algae are one of three types of algae, i.e., brown algae (Phaeophyta), red algae (Rhodophyta) and green algae (Chlorophyta) classified based on their color and major photosynthetic pigments. Brown algae contain chlorophyll a, chlorophyll c and fucoxanthin, red algae contain chlorophyll a, chlorophyll d and phycoerythrin, while green algae contain chlorophyll a, chlorophyll b and xanthophylls. Due to the different abiotic and biotic factors in the marine environment and their distinct evolutionary origin seaweeds are a rich source of unique compounds of which several have demonstrated health benefits. Bioactive compounds of interest found in brown seaweed include polysaccharides (e.g., alginate, fucoidan), proteins (e.g., phycobiliproteins), polyphenols (e.g., phlorotannins), carotenoids (e.g., fucoxanthin), phytosterols (fucosterol) and *n*-3 long-chain polyunsaturated fatty acids (e.g., eicosapentaenoic acid) [1]. They have been reported to have beneficial effects in various diseases, including metabolic diseases, diabetes mellitus, cardiovascular disease, cancer and neurodegenerative diseases.

We structure this review according to the distinct compounds in brown seaweed that have been applied, and therefore distinguish effects of whole seaweeds or crude extracts (described in Section 3), polysaccharides such as fucoidan and alginate (Section 4), compounds with ring-shaped structures such as phytosterols (e.g., fucosterol) and (poly)phenols (e.g., phloroglucinol and phlorotannins) (Section 5) and carotenoids, in particular fucoxanthin, fucoxanthinol and meroterpenoids (Section 6).

For each of these compounds we discuss the demonstrated effects on different phases of the inflammatory response (schematically depicted in Figure 1). We describe (i) studies that investigate the brown seaweed (compound) effects on steady state immune parameters that influence subsequent responses to inflammatory challenges. Then, in view of the focus of this special issue on allergic disease, we separately discuss (ii) studies aimed at identifying the effects of intake of brown seaweed or -components on allergies and models thereof. As next steps (iii) and (iv), we distinguish brown seaweed effects on different phases in other acute inflammatory responses, with their characteristic read-outs as indicated in Table 1. Briefly, here we distinguish the first, immediate response (iii) typified by reactive oxygen radical production and release of early mediators such as IL-1, TNF and arachidonic acid metabolites. Then, we consider (iv) the seaweed effects on the second inflammation phase characterized by enhanced production of above-mentioned cytokines as well as secondary cytokines, such as IL-10, chemokines and other mediators, and infiltration of leukocytes in tissue. Finally, we discuss (v) the effects of seaweed intake on chronic inflammation, which is mostly low-grade, and their sequels. These include insulin-, leptin- or glucocorticoid resistance, which are related to the induction of diabetes, obesity and hampered down-regulation of inflammation, respectively. Furthermore, malignant transformation may be another consequence of chronic exposure to inflammatory conditions.

Box 1Brief introduction on the inflammatory response.Inflammation is the response of tissues to any trigger significantly disturbing homeostatic conditions. These triggers can be manifold, and vary from infection or trauma, to metabolic challenges. The response is initiated by recognition of damage- or danger-related molecules (so-called DAMPs) that become exposed and are sensed by parenchymal tissue cells and resident innate immune cells, in particular macrophages and mast cells.Activation of these cells (Phase 1) stimulates a cascade of events, within the first seconds to minutes, involving local release of ready-made mediators such as histamine, inflammatory arachidonic acid products such as PGE_2_, and stored cytokines. Oxygen radicals (reactive oxygen species; ROS), produced by NADPH-oxidase complex assembled upon cellular activation and by mitochondria, further amplify activation of resident cells. In addition to their essential role in redox signaling, ROS are important as microbicidal molecules and as inducers of oxidative damage. Secreted first wave mediators alarm neighboring cells and stimulate local vascular cell activation, causing upregulation of adhesion molecules on the endothelial cell surface, and vasodilation. This enables plasma fluid, including bioactive proteins, to penetrate the tissue.Subsequently, activated resident cells produce inflammatory cytokines and chemokines by de novo gene transcription (Phase 2). Inflammatory signaling pathways involving NF-κB and AP-1, triggered by the receptor-DAMP interaction, are essential in this process. The released mediators amplify the inflammatory response and enable the recruitment of leukocytes from the circulation. Based on the profile of released chemokines, neutrophils are the most numerous attracted cells in the initial response. In addition, expression of inducible nitric oxide synthase (iNOS) is stimulated by bacterial products and pro-inflammatory cytokines, in particular IFN-γ. iNOS generates nitric oxide (NO), which is an important inflammatory mediator as microbicidal product of especially activated macrophages, and as an autocrine and paracrine signaling molecule. In this second phase of inflammation, also anti-inflammatory mediators are produced, which include arachidonic products such as lipoxins and resolvins, and cytokines such as IL-10. Dependent on the strength of the inflammatory trigger, systemic responses may occur, including activation of the hypothalamus-pituitary-adrenal (HPA) axis, leading to cortisol or corticosterone release. The first systemic cytokine wave initiates the production of so-called acute phase proteins such as C-reactive protein (CRP) from the liver, which occurs from approximately 24 h after the initial trigger. An adaptive immune response is initiated as DAMPs and -related antigens are transported to draining lymph nodes via lymph and migrating antigen-presenting cells. This leads to proliferation of antigen-specific B- and T-lymphocytes that eventually leave the lymph nodes and become effective in the periphery from approximately 4 days after initiation.Dependent on the nature and severity of the initial trigger, and ability to annihilate it, repair processes are initiated after a few days. If the trigger remains, such as in case of adverse metabolic conditions or persistent infection, inflammation may become chronic. Then a new equilibrium is sought (Phase 3), which affects tissue performance and the response to further environmental triggers. Long-term exposure to adverse conditions thus may lead for instance to resistance to regulating hormones, such as insulin or cortisol.Allergic responses are inflammatory responses triggered by pre-existing antibodies or primed T-lymphocytes specific for harmless molecules from the environment. Best known are the responses caused by activation of mast cells evoked by IgE antibodies specific for innocuous antigens such as house dust mite or pollen. Other antibody subclasses with adverse specificities, for instance against certain drugs, also can cause clinical responses. Finally, T-lymphocytes with unfavorable specificities may cause so-called delayed-type hypersensitivity responses. The delay is explained by the necessity of T-lymphocytes to migrate to the site of antigen exposure and presentation, and to become activated locally. In essence, allergic responses follow the phases as outlined for the inflammatory response in general.

Challenges to generating a comprehensible overview are the multifaceted aspects of allergy and inflammation, besides the large variety of brown seaweed species and preparations, as well as the different host species in which seaweed administration has been tested. Furthermore, this divergency bears a risk of overstretching conclusions based on limited findings. With these caveats in mind, we attempt to describe the commonalities between different studies, or discuss specific findings that deserve attention for future development. Detailed findings from the individual publications have been summarized in tables, linked to the compound categories mentioned above.

## 2. Approach to Systematic Search

The aim of this systematic review is to investigate the effect of brown seaweed components as a dietary supplement on inflammation with extra focus on allergies.

### 2.1. Databases and Search Strategy

We performed two searches, in May and in November 2020. Several electronic databases were used to include studies: Embase via Embase.com (1971–Present), Medline ALL via Ovid (1946–Present), Web of Science Core Collection via Web of Knowledge (1975–Present), Cochrane Central Register of Controlled Trials via Wiley (1992–Present) and Google Scholar. References were filtered for duplicates in Endnote. The systematic search was executed by Elise Krabbendam, Biomedical Information Specialist at the Medical Library of Erasmus MC. The exact search terms are shown in the Appendix A, where also a detailed flow chart of paper selection is shown. First, at least two review authors independently assessed title and abstract of the articles based on inclusion and exclusion criteria mentioned below. This was carried out using Endnote X9 software, based on reviewing methods described by Bramer et al. 2017 [2]. Secondly, after independent assessment of the articles, references selected by both review authors were included in a preliminary database. For final full text inclusion all references were combined and assessed by all review authors to assure eligibility and to extract important information to be included in the tables. References were divided by seaweed component among all review authors and based on full text selected to be included or excluded from the systematic review. Third, some relevant articles were included that were not retrieved in the systematic search, primarily because searchable parts of the articles did not contain reference to terms related to inflammation.

### 2.2. Eligibility Criteria

For this systematic review we only included articles written in English exclusively. Review articles were excluded from the search. The aim of the first search was to include papers addressing the effect of brown seaweed as nutritional supplement on leucocytes and how this contributes to the inflammatory process. This resulted in a total of 906 references after deduplication. For extra focus on allergy and atopy the search was expanded, resulting in a final total of 1035 references after deduplication. Papers on components of brown seaweed or whole extracts were included, but only when orally administered to human or animals. Exclusion criteria for this systematic review were seaweeds other than brown species or no oral administered. Furthermore, articles mentioning fucoidan as control inhibitor/blocker for selectins or scavenger receptor exclusively, were also excluded.

## 3. Whole Seaweed or Crude Extract Supplementation

### 3.1. Whole Seaweed or Crude Extract: Effects in Steady State

Table A1 summarizes the main findings of oral supplementation of whole seaweeds or crude extracts on inflammation-related parameters. Safety aspects of seaweed consumption are important to consider. Potential overload with iodine or heavy metals are major long-term risks when unprocessed edible seaweeds are ingested [1]. Short-term monitoring after administration of single doses of crude extracts in rodents showed no effects of toxicity up to 5000 mg/kg in mice (*Sargassum micracanthum* [3]; *Cystoseira compressa* (Esper) [4]), or showed LD50 values of 1000–2000 mg/kg (*Fucus vesiculosis* [5]). Considering a dosage of 200 mg/kg/day is routinely applied for regular use, it may be argued that the safe dosage range might be limited. However, specific toxicity may be highly batch-dependent and related to toxic contaminants rather than seaweed content.

A widely studied facet of seaweed supplementation is its high anti-oxidant activity, and related to this, anti-inflammatory activity. Human studies in this direction are scarce, however. Consumption of 4.8 g dried *Sargassum muticum* per day for a period of 4 weeks by healthy volunteers stimulated an increased total anti-oxidant status in serum, correlated to decreased concentrations of oxidized LDL [6]. In contrast to general assumptions, Baldrick et al. observed no significant changes in oxidative or inflammatory parameters after oral consumption of *Ascophyllum nodosum* extract by individuals with overweight or obesity (100 mg/day, 8 weeks) [7]. An interesting aspect of the latter study is that individuals differed notably (up to >4000×) in the total amount of seaweed polyphenol metabolites present in urine. It is tempting to speculate this might be attributed to differences in microbiota composition between individuals. In a study in goat, where *Ascophyllum nodosum* extract was added (2%) to feed, an increased anti-oxidant status was shown [8]. Similarly, 4-week treatment of rats with *Fucus vesiculosus* extract stimulated increased serum paraoxonase and superoxide dismutase activities, thus leading to an increased anti-oxidant status [5].

Effects of seaweed supplementation on blood cell counts are variable; studies in human or other mammals showed limited effects [6,8], but increased counts were observed when chicken and fish were fed *Laminaria japonica* or *Sargassum oligocystum*, respectively [9,10]. Such addition to animal feed is not only associated with increased growth and feed conversion ratios in chicken and fish, but also enhanced status of innate and adaptive immune defenses and immune responsiveness and survival after infectious challenge [11,12,13].

### 3.2. Whole Seaweed or Crude Extract: Effects on Allergy 

Importantly, oral brown seaweed supplementation shows consistent beneficial effects in different models of allergy (Table A1). *Eisenia* (=*Ecklonia*) *arborea* powder intake by Brown Norway rats, orally sensitized to ovalbumin, leads to decreased serum IgE and histamine levels and decreased IL-4 and IL-10 production in lymphoid organs, while IFN-γ synthesis is increased. This indicates a favorable change in Th1/Th2 balance towards the former [14].

Mouse models using 2,4-dinitrophenol (DNP) or 2,4-dinitrochlorobenzene (DNCB) sensibilization and challenge are much used in this field, and similarly show positive effects of brown seaweed intake. *Sargassum horneri* extract has anti-allergic activity by suppressing degranulation of mast cells and basophils. This reduces nasal rubbing [15] or clinical signs of atopic dermatitis, as well as inflammatory cytokine levels and leukocyte skin infiltrates [16] in these allergy models. Similarly, application of extracts from *Costaria costata* [17] and *Laminaria japonica* [18] reduce severity of allergic dermatitis and stimulate healing, possibly by decreasing inflammatory pathways in keratinocytes.

### 3.3. Whole Seaweed or Crude Extract: Effects in Acute and Chronic Inflammation

Without prior sensitization, irritant application on murine skin also causes signs of acute inflammation associated with local mast cell degranulation and increased vascular permeability. Feeding rats with *Laminaria japonica* extract decreases paw swelling and leukocyte infiltration induced by carrageenan application, likely by inhibiting NF-κB activation causing decreased inflammatory mediator production [19] (Table A1). Similarly, oral or topical administration of *Sargassum fusiforme* extract significantly reduces ear swelling by inhibiting mast cell degranulation and enzymes involved in production of inflammatory arachidonic acid mediators [20].

The challenge of experimental animals or isolated cells with lipopolysaccharide (LPS) is a common model of acute inflammation, mimicking the response to bacterial infection. A 5-week treatment of rats with *Eisenia* (=*Ecklonia*) *bicyclis* extract mediated a reduced inflammatory activation of peritoneal macrophages upon in vitro LPS stimulation through inhibition of NF-κB activity [21]. This was associated with reduced iNOS expression and nitric oxide (NO) production. In marked contrast, a similar study in mice showed that 3-week oral treatment with *Sargassum fusiforme* (also called *Hizikia*) extract slightly potentiated the production of IL-1β, IL-6 and TNF-α by isolated peritoneal macrophages stimulated in vitro with LPS [22]. In accordance with an inflammation-regulating effect, 4-day oral application of *Sargassum serratifolium* extract in mice caused significantly reduced production of TNF-α, IL-1β and IL-6 upon in vivo LPS challenge [23]. This confirmed the in vitro findings of direct inhibition of NF-κB activation and nuclear translocation by seaweed components.

Investigation of seaweed treatment on LPS responses in other than murine species generally corroborated an inflammation-inhibiting effect. Challenge of zebrafish embryos with LPS or H_2_O_2_ showed reduced reactive oxygen species (ROS) production and associated cell death when treated simultaneously with *Sargassum polycystum* or *Chnoospora minima* extract [24].

Adding *Sargassum latifolium* to sheep feed caused a reduced inflammatory response to LPS challenge and increased blood anti-oxidant defense capacity [25]. This seaweed treatment also mediated a reduced inflammatory response to heat stress challenge in these sheep [26]. The latter study confirms earlier work in lamb, showing reduction of heat stress-related effects on leukocyte oxidative and phagocytic function by *Ascophyllum nodosum* extract administration [27].

Furthermore, in mouse models of inflammatory disease, seaweed supplementation has shown beneficial effects. In dextran sulfate sodium-induced chronic colitis, application of *Turbinaria ornata* extract causes decreased disease activity as indicated by colon length, histomorphological index and myeloperoxidase activity [28]. This was accompanied by increased expression of regulatory T-cell-associated FoxP3 and anti-inflammatory IL-10. Similarly, *Laminaria japonica* extract caused a significant diminution of colitis signs in this model, and simultaneous application of bacterial probiotics showed synergistic beneficial effects on histological score and decreased levels of some proinflammatory cytokines [29]. In a mouse model of arthritis induced by bovine type II collagen immunization, oral supplementation with *Sargassum muticum* extract significantly decreased the arthritis and edema scores as well as TNF, IL-6 and IFN-γ levels [30].

In one of the scarce human studies, Cooper et al. found that individuals with active Herpes infection showed increased healing rates with *Undaria pinnatifida* consumption, while latent Herpes carriers did not experience viral reactivation [31]. Investigating the mechanisms, the authors found the extract strongly inhibited Herpes virus infectivity in vitro, and stimulated human T cell mitogenesis, thus potentiating adaptive immune responses.

A study using a phylogenetically more distinct organism, kuruma shrimp, indicated that oral supplementation with *Laminaria japonica* significantly increased survival upon White Spot Syndrome virus infection [32]. This was accompanied by enhancement of chemotaxis as well as other defense mechanisms by hemolymph leukocytes (hemocytes), including superoxide production and antioxidative phenoloxidase activity upon appropriate stimulation.

### 3.4. Whole Seaweed or Crude Extract: Late Consequences of Inflammation and Sequels 

Acute or chronic inflammatory conditions influence local and systemic tissue responses, and thus seaweed supplementation also affects peripheral tissue function in inflammation (Table A1). In a rat model of ligature-induced periodontitis, *Sargassum fusiforme (Hizikia)* application reduced alveolar bone loss, related to decreased osteoclast and increased osteoblast gene expression in vitro [33]. Furthermore, in a model of autoimmune thyroiditis, a traditional Chinese medicine combination of 10 different herbs, including *Sargassum fusiforme*, mediated a decrease in autoimmune thyroiditis and anti-thyroid autoantibody formation [34]. Omission of *Sargassum fusiforme* in this model significantly diminished the protective effect.

In recent years, the link between adipose tissue metabolic dysregulation and inflammation has been recognized increasingly [35]. Several studies investigated metabolic effects of seaweed application, in particular *Undaria pinnatifida*, in murine models of obesity and type 2 diabetes induced by a high-fat diet [36,37,38]. Seaweed was in some studies combined with other nutraceuticals. In general, the obesity phenotype did not change, while improvement of glucose regulation was only observed by Oh et al. (2016) [36], but not in the other two studies. Other aspects, however, showed beneficial effects of seaweed supplementation, such as microbiome composition, MCP-1 induction [38], systolic blood pressure and non-esterified fatty acid levels [37] or presence of clusters of necrotic adipocytes surrounded by macrophages in adipose tissue (so-called crown-like structures) [36].

A pathological condition strongly related to obesity is the polycystic ovary syndrome (PCOS). In a rat model of PCOS, application of *Ecklonia cava* extract mediated a decrease in vaginal leukocyte infiltration, and restored hormonal levels and irregular ovarian cycles [39]. However, it did not inhibit the weight gain associated with PCOS induction.

## 4. Brown Seaweed Polysaccharide Supplementation 

Among polysaccharides present in brown seaweed fucoidan has received most attention as a constituent with diverse bioactive effects. Furthermore, bioactivity has been demonstrated of the polysaccharides laminarin, a beta-glucan polysaccharide and alginate, a linear acidic soluble dietary polysaccharide.

Fucoidans are a group of polysaccharides (fucans) primarily composed of sulphated L-fucose with less than 10% of other monosaccharides. They are widely found in the cell walls of brown seaweeds, but not in other algae or higher terrestrial plants [40]. The major function of fucoidans in cell walls is mechanical support and protection against desiccation during air-exposure of the seaweed at low tide. The amount of fucoidan in brown seaweeds is variable; 8–20% of dry weight with the highest content of about 20% being detected in *Fucus*
*vesiculosus* [41,42].

A number of health-improving effects have been ascribed to fucoidans [40,41,43]. Biological activities of fucoidans, such as antioxidant and anti-coagulant capacity, are affected by their molecular weight and sulphated ester content, both the number of sulphate groups, determining the negative charge of the molecule and the position of the sulphate groups on the sugar residues [40,44]. The biological activity of fucoidan is also affected by the glucuronic acid and fucose content. The molecular weight of fucoidan ranges from for example from 50 to 80 KDa in *Undaria*
*pinnatifida* and *Fucus*
*vesiculosus,* respectively, to 1920 kDa in *Cladosiphon* species [45], with multiple sizes being present in certain species. Low molecular weight (LMW) fucoidan is produced by enzymatic digestion or acid hydrolysis of naturally occurring high molecular weight (HWM) fucoidan. Application of different molecular species of fucoidan obtained by different methods of purification and treatments such as hydrolysis complicates interpretation of results.

### 4.1. Brown Seaweed Polysaccharide Effects in Steady State

Fucoidan is absorbed in limited amounts in the gastrointestinal tract after oral intake [46,47]. In Japanese populations where brown seaweed is part of daily diet, systemic fucoidan uptake was shown by its presence in serum and urine [48]. Protective effects of fucoidan on the intestinal epithelial barrier function were observed in vitro. Fucoidan protected the tight junctions from oxidative injury and upregulated the expression of claudin-1 [49]. Table A2 summarizes the effects of fucoidan on different aspects of inflammation.

Fucoidan is not toxic, but high dosages can induce increased bleeding time. In rats no toxicity as observed after oral administration of a single dose of *Ascophyllum nodosum* fucoidan of 2000 mg/kg [50] or 300 mg/kg/day *Laminaria japonica* fucoidan for 6 months [51]. However, application of a daily dose of 2500 mg/kg for 6 months resulted in increased bleeding time. The application of fucoidan in food has been approved for human consumption up to 250 mg/day by the European Food Safety Authority, EFSA [1].

### 4.2. Effects of Polysaccharides from Brown Seaweed in Allergy, Acute Inflammation and in the Modulation of Immune Responses

Both pro- and anti-inflammatory effects of fucoidan have been reported. In macrophages fucoidan treatment induced NF-κB nuclear translocation, followed by iNOS and COX-2 transcription, inducing the secretion of the pro-inflammatory cytokines IFN-γ, TNF-α and IL-1β and of inflammatory mediators NO and PGE2 [52]. However, pre-treatment of macrophages and lymphocytes with fucoidan prior to stimulation with LPS or other pro-inflammatory stimuli was found to blunt the pro-inflammatory reaction or induces an anti-inflammatory effect, resulting in inhibition of NF-κB translocation and in lower levels of pro-inflammatory mediator production [53,54,55,56].

Below an elaboration on anti-allergy effects and enhanced immune effects in production animals and innate and adaptive immune system modulation studies in mice is described.

#### 4.2.1. Anti-Allergic Effects of Brown Seaweed-Derived Polysaccharides

Overall, oral supplementation of brown seaweed polysaccharides was reported to inhibit allergic responses via multiple mechanisms. The polysaccharides were shown to be an effective agent antagonizing IgE production as examined in different ovalbumin (OVA)-sensitized mouse models [57,58], but also an allergy-specific mechanism of oral fucoidan supplementation has been found in its capacity to induce galectin-9 production from intestinal cells [59,60]. Galectin-9, belonging to a soluble lectin family, recognizes β-galactoside and prevents IgE binding to mast cells, consequently inhibiting mast cell degranulation. Accordingly, fucoidan from *Saccharina japonica* (400 μg for 4 days) was found to increase circulating galectin-9 [59]. After OVA-immunization the allergic symptoms in sensitized mice were reduced by fucoidan (60 μg/mouse/d for 17 days) via inducing galectin-9 production from colonic epithelial cells [60].

In several OVA-immunized mouse models, oral administration of fucoidan or a polysaccharide fraction was shown to have anti-allergy activity. Application of a polysaccharide fraction from *Laminaria japonica* (50 mg/kg/day for 2 weeks) in a mouse model of asthma significantly decreased the numbers of eosinophils in the bronchoalveolar fluid and alleviated lung inflammation compared to the non-treated control mice [58]. It also reduced serum IgE concentrations and decreased the concentrations of IL-13 and TGF-β1 in bronchoalveolar fluid and expression in lung, while increasing expression of IL-12. Similarly, *Laminaria japonica* fucoidan ingestion (200, 600, 1000 mg/kg for 6 weeks) decreased OVA-specific IgE in mice [61]. Fucoidan from *Undaria*
*pinnatifida* (400 mg/kg for 7 days) inhibited particulate matter-induced exacerbation of allergic asthma [57]. Specifically, fucoidan treatment significantly attenuated lipid peroxidation, infiltration of inflammatory cells and Th2-related IL-4 concentrations. Furthermore, fucoidan suppressed mast cell activation, degranulation and IgE synthesis as well as mucus hypersecretion and goblet cell hyperplasia. This also is reflected in immunoglobulin isotypes produced as *Cladosiphon*-fucoidan dose-dependently (up to 1025 mg/kg body weight for 8 weeks) increased systemic IgM, IgG and IgA levels, while decreasing IgE and IL-4 significantly [62].

The observed changes are suggestive of a shift from Th2 to Th1 induced by orally ingested fucoidan. Enhanced IL-12 and IFN-γ production by ingestion of *Tetragenococcus halophilus* KK221, a probiotic known for its anti-allergic properties, was even further increased by combined ingestion of the probiotic and LMW fucoidan isolated from *Undaria pinnatifida* in OVA-immunized mice. The results indicated an extra shift towards Th1.

Furthermore, alginate was found to improve (food) allergy outcomes in an OVA-sensitized mouse model [63]. Ingestion of alginate (2 mg) extracted from *Laminaria japonica* one day before oral application of ovalbumin improved integrity of intestinal epithelial villi and inhibited mast cell degranulation in the jejunum. Serum levels of IgE, histamine and IL-4 were significantly lower, while IFN-γ was markedly increased. Furthermore, Tregs in spleen were increased, while OVA-induced differentiation of Th0 cells into Th2 cells was inhibited [63].

Overall, brown seaweed-derived polysaccharides generally appear to modulate the Th1/Th2 balance and mast cell degranulation in favor of an anti-allergic effect. This shows that fucoidan is potentially an effective therapeutic agent for type I allergic diseases.

#### 4.2.2. Effects of Brown Seaweed Polysaccharides on Innate and Adaptive Immune System (Production Animals)

In search for alternatives for antibiotics in production animals, brown seaweed polysaccharides and especially fucoidan, have appeared as promising functional feed additives. Fucoidan and laminarin were found to improve the immune response of pregnant sows and piglets prior to or while suckling [64], and after weaning [65].

Dietary supplementation of sows in the final part of gestation with *Laminaria* spp. extract increased IgG and IgA in sow colostrum by 19% to 25% [64]. Consequently, also a 10% increase in piglet serum IgG was observed. This suggests an important effect of maternal diet on the immune status of piglets. Dietary supplementation with an extract of *Ascophyllum nodosum* and *Fucus* in sows (30 g/day from the 85th day of gestation until weaning) resulted in an increased population of CD4+CD8+ T cells in the thymus, spleen, mesenteric node, liver and in peripheral blood as compared to the control group [66]. Piglets from laminarin-fed sows (1.0 g/d from day 107 of gestation until weaning) showed down-regulation of IL-6 mRNA expression in the colon at weaning and of IL-8 in the ileum on day 8 post weaning compared with those from the non-laminarin-fed sows [67].

Weaning of piglets is a stressful event for piglets and is often associated with pro-inflammatory immune effects in the piglets’ gastro-intestinal tract. Addition of laminarin to weaning piglets’ diets resulted in lower expression of pro-inflammatory cytokines IL-1β, IL-6 and IL-17 in colonic mucosa [65]. Even though these positive effects were observed in piglets, laminarin did not result in any detectable benefits in Friesian bull calves [68].

In Salmonella-challenged broiler chickens, addition of 0.2% alginate oligosaccharides to the regular diet inhibited *Salmonella*
*enteritis* colonization, possibly by increasing colonic anti-Salmonella IgA levels [69]. In unchallenged broiler chickens, supplementation of 0.2% alginate oligosaccharides showed dramatic immunostimulatory activity by inducing interferon-γ, IL-10 and IL-1β mRNA expression in cecal tonsils. Interestingly, the robust mucosal immune response in the absence of a challenge was related to a decline in body weight, as compared to the control group.

In aquaculture, several studies pointed at improved innate immune markers upon fucoidan and laminarin supplementation in shrimp and fish [70,71,72,73,74,75,76]. In addition, higher survival rates during bacterial challenges were observed in the supplemented animals, as compared with those fed a regular diet.

Taken together, enhancing the innate and adaptive immune system by oral ingestion of seaweed-derived polysaccharides and oligosaccharides is a promising solution for improving animal health, reducing infection incidence and reducing the need for antibiotics use.

#### 4.2.3. Effects of Brown Seaweed Polysaccharides on Innate and Adaptive Immune System (Mouse Models)

Polysaccharides obtained from brown seaweed may support various aspects of the immune system in both immunocompetent and immunosuppressed states. For instance, oral ingestion of a polysaccharide extract from *Kjellmaniella crassifolia* (2 weeks) by C57BL/6 mice, resulted in enhanced IFN-γ, IL-12, IL-6 and IgA secretions by spleen cell cultures upon concanavalin-A stimulation [77]. Orally administered LMW fucoidan (200–1000 mg/kg for 6 weeks) from *Laminaria japonica* to BALB/c mice stimulated the innate immune system by increasing natural killer (NK) cell activity and peritoneal macrophage phagocytic activity [61]. LMW fucoidan also increased IL-2, IL-4 and IFN-γ secretion by splenocytes and IgG and IgA concentrations in serum, while it decreased OVA-specific IgE. In bacterial antigen-stimulated immune responses, the IgM and IgG concentrations in serum were significantly higher in the LMW fucoidan group than in the control group. In addition, an LMW fucoidan-enriched extract from *Okinawa mozuku* orally administered (up to 1025.0 mg/kg for 6 weeks) to BALB/c mice resulted in enhanced splenocyte proliferation and secreted IL-2 levels, as well as in increased macrophage phagocytic activity, and serum IgM, IgG and IgA, while splenocyte-secreted IL-4 and IL-5 were decreased, and also serum IgE was decreased significantly [62]. Interestingly, HMW fucoidan (50 g/kg) but not LMW and IMW fucoidan, increased the relative number of cytotoxic T-cells in spleens of Balb/c mice [78]. These immune-potentiating effects appear to be effective in infection as complete elimination of liver and spleen parasite burden was achieved by fucoidan (200 mg/kg, 3 times weekly, for 6-weeks) in a mouse model of *Leishmania donovanii* infection [79]. This curative effect was associated with switching of T cell differentiation from Th2 to Th1 mode.

In addition to its capacity to enhance the innate and adaptive immune system, oral fucoidan is an interesting candidate for antiviral therapies related to its intrinsic capacity as a competitive binding agent for envelope viruses, thus preventing cellular entrance [80]. Oral ingestion of fucoidan improves the outcome in virus-infection mouse models with respect to viral load [81], serum antibody levels and overall survival [82,83] in immunocompetent and immune-suppressed mice. Furthermore, fucoidan extracted from *Undaria pinnatifida* protected both immunocompetent and immunosuppressed mice from infection with HSV-1 as indicated by the improved survival rate and lesion scores [82]. In immunocompetent mice fucoidan enhanced activity of CTL and increased circulating anti-HSV antibodies in HSV-1-infected mice.

In an immunosuppressed state, selective augmentation of NK activity was observed upon oral treatment with *Undaria* fucoidan, but this induced no significant change in NK activity in immunocompetent mice where a normal level of NK activity was maintained. Fucoidan extracted from *Undaria pinnatifida* showed also beneficial effects during influenza virus infection in immunocompetent an immunosuppressed mice [83]. Fucoidan administration (7 days prior to virus inoculation until 7 days after inoculation (2 × 5 mg/day)) resulted in significant increase in neutralizing antibody titers in bronchoalveolar lavage fluids in both healthy mice and mice with suppressed immunity as compared with placebo groups.

Furthermore, in the defense against tumors fucoidans enhance innate and adaptive immune responses. Different fucoidans were found to increase immune reactions in various tumor models, leading to comparable or even better results than standard chemotherapy exclusively [84,85,86,87]. Consumption of fucoidan isolated from *Undaria pinnatifida* (1% of the diet for a period of 10 days) showed tumor inhibition in an A20 leukemia mouse model, related to enhanced Th1 and NK cell activity [84]. Oral intake of polysaccharides from *Sargassum fusiforme* (100 and 200 mg/kg for 28 days) significantly inhibited the growth of A549 lung adenocarcinoma in mice, but also remarkably promoted IL-1 and TNF-α production from peritoneal macrophages, increased serum TNF-α levels and splenocyte proliferation [87]. Oral administration of fucoidan extracted from *Cladosiphon okamuranus* (5 g/kg/day for 28 days) also inhibited tumor growth and increased survival time in a colon 26 tumor-bearing mouse model. In the spleens of these mice, an increased population of NK cells was observed. Furthermore, in an experiment applying the same fucoidan to MyD88 knockout mice, a model for investigating TLR4 signaling pathways, it was found that the observed anti-tumor effects are related to gut immunity [85]. Furthermore, polysaccharide extract from *Sargassum fusiforme* (400 mg/kg for 28 days) exerted anti-tumor and immunomodulatory activities in nasopharyngeal carcinoma [86] and hepatic carcinoma tumor-bearing mice [88]. In a xenograft tumor model orally administered fucoidan from *Fucus vesiculosus* (150 mg/kg body weight daily for 2 weeks) increased cytolytic activity of NK cells and significantly delayed tumor growth [89]. Furthermore, in a rat model for experimental mammary carcinogenesis administration of fucoidan (400 mg/kg/day for 4 months) showed protective and immunomodulatory effects [90]. Tumor growth in Sarcoma 180 (S-180)-bearing mice was delayed by ingestion of fucoidan from *Cladosiphon okamuranus*, which stimulated NO production by macrophages via NF-κB-dependent signaling pathways [52]. Oral intake of ascophyllan, a sulphated polysaccharide obtained from *Ascophyllum nodosum,* (50 and 500 mg/kg), also delayed tumor growth. Interestingly, oral ingestion significantly increased serum IL-12 and TNF-α levels and mediated better overall outcome compared to intraperitoneal application in S-180 mice, where immune markers did not change [91].

Seaweed polysaccharides can also function as immune-stimulating adjuvant in immunosuppressed states during chemotherapy. Oral intake of polysaccharide extract from *Sargassum fusiforme* (200 mg/kg for 6 days) was identified as a potent immune-enhancing agent in immunosuppressed mice [92]. Oral administration of fucoidan (150 mg/kg for 14 days) resulted in enhanced recovery of all T cell populations (CD3+, CD4+, CD8+) and of the proliferative capacity of splenocytes in immunosuppressed mice [93]. Furthermore, laminarin administration (500–1000 mg/kg/day for 10 days) induced IL-12 and IFN-γ in immunosuppressed mice [94]. Taken together, oral intake of brown seaweed polysaccharides is shown to be an effective immune enhancer in a wide variety of mouse models.

#### 4.2.4. Anti-Inflammatory Effects of Fucoidan in Animal Models and Clinical Trials

Fucoidans extracted from different seaweed species and molecular sizes showed anti-inflammatory effect in a wide range of acute and chronic inflammation models in mice.

In an arachidonic acid-induced ear inflammation model sulphated polysaccharide extracted from *Sargassum*
*hemiphyllum* decreased ear swelling and erythema [95]. The polysaccharides decreased the local levels of myeloperoxidase, nitric oxide, IL-1β, IL-6 and TNF-α in a dose-dependent manner (20–80 mg/kg body weight for 5 consecutive days). Histological examination revealed that the polysaccharides reduced the area of neutrophilic infiltration in inflamed ears. Similarly, oral ingestion of fucoidan extracted from *Undaria*
*pinnatifida* (0.5 mg for 20 days) inhibited the inflammatory reaction in a mouse model where LPS was injected buccally [55]. In the same set up, but now using bacterial infection, fucoidan reduced inflammation but did not lead to clearance of the bacterial infection, nor to prevention of infection-related bone loss. In a carrageenan-induced air pouch inflammation model, a preparation of fucoidan (54 mg/kg for 7 days) inhibited inflammatory markers and showed reduced attraction of inflammatory cells as demonstrated by histology [96]. Pretreatment with orally administered fucoidan (20 mg/kg for 2 weeks) reduced mucosal lining inflammation and prevented elevation of serum IL-6 levels, while levels of serum IL-10 increased in an aspirin-induced mucosal ulcer model in mice [97]. Accordingly, *Cladosiphon* fucoidan (chow containing 0.05% *w*/*w*) ingestion beneficially affected murine dextran sulphate sodium (DSS)-induced colitis [98]. The lamina propria of inflamed colon showed reduced amounts of IL-6 and IFN-γ, and an increase in IL-10 and TGF-β upon fucoidan treatment. Murine DSS-induced colitis significantly improved upon treatment with fucoidan (10 mg/day for 1 week) [99]. Treatment with a fucoidan-polyphenol complex showed even better results as it reduced IL-12, TNF-α and IL-6 in colitis tissue and ameliorated colitis-related visible body markers, such as weight loss and blood in stool. In contrast, when this complex was injected intraperitoneally, it was unable to reduce disease severity and even deteriorated some colitis markers. In mice, colonic inflammation and microbiota dysbiosis induced by antibiotics was alleviated by administration of fucoidan extracted from *Ascophyllum nodosum* (400 mg/kg for 28 days) [100]. Fucoidan prevented colon shortening and colon tissue damage, and it improved abundance of beneficial microbes while decreasing harmful microbes. In a model with chemically induced colorectal cancer, ingestion of *Fucus*
*vesiculosus* fucoidan was shown to protect against tumorigenesis and to reduce colorectal inflammation and dysbiosis [101].

The bioactivity of fucoidan is related to its molecular weight. LMW and HMW fucoidan from *Undaria pinnatifada* were tested in a murine model of collagen-induced arthritis [102]. LMW fucoidan protected against tissue degeneration, while the same dose of HMW fucoidan worsened it. In accordance, LMWF reduced the severity of arthritis and the levels of Th1-dependent collagen-specific IgG2a, while HMWF enhanced the severity arthritis and the levels of collagen-specific antibodies.

Furthermore, in different acute inflammation models in zebrafish embryos, strikingly similar anti-inflammatory effects were noticed [53,54,56,103]. Administration of fucoidan (25–100 µg/mL) one hour prior to LPS treatment improved survival of zebrafish embryos and diminished inflammatory markers.

In support of an inflammation-regulating effect of fucoidan, oral administration of *Laminaria japonica* fucoidan (50–200 mg/kg) protected against myocardial ischemia–reperfusion injury in rats in a dose-dependent manner [104]. Furthermore, oral ingestion of enzymatically hydrolyzed fucoidan extracted from *Sargassum*
*hemiphyllum* (200 mg/kg/day for 14 days) decreased radiation-induced pneumonitis and lung fibrosis by reducing inflammatory cytokine expression in lung tissues [105]. In both models decreased accumulation of neutrophils and macrophages was observed.

In a mouse model of chronic infection with *Schistosoma japonicum* oral ingestion of *Fucus vesiculosus* fucoidan (500 mg/kg per 2 days for 40 days) significantly reduced the hepatic granuloma size and fibrosis response [106]. Lower levels of pro-inflammatory cytokines were observed in the livers of fucoidan-treated infected mice. Infiltration of Treg cells and levels of IL-10 and TGF-β were significantly enhanced in both the livers and spleens from fucoidan-treated mice. Another study aimed to explore the effects of fucoidan from *Fucus vesiculosus* on concanavalin A (ConA)-induced acute liver injury in mice. Pretreatment with fucoidan (10–50 mg/kg for 2 weeks) protected liver function indicated by ALT, AST and histopathological changes by suppressing inflammatory cytokines, TNF-α and IFN-γ [107]. The results demonstrated that fucoidan alleviated ConA-induced acute liver injury via the inhibition of intrinsic and extrinsic apoptosis mediated by the TRADD/TRAF2 and JAK2/STAT1 pathways which were activated by TNF-α and IFN-γ.

Fucoidan from *Fucus vesiculosis* (300–600 mg/kg) was shown to be able to delay the onset and incidence of autoimmune diabetes in non-obese diabetic mice via regulating DC/Treg-induced immune tolerance via induction of IL-10 and TGF-β, while reducing the levels of IL-6 and IFN-γ [108]. In the pancreas TLR4 expression and the downstream molecules were downregulated while pancreatic internal environment was maintained in the fucoidan-treated groups.

In a clinical trial in patients with chronic hepatitis B infection, oral ingestion of a commercial oligo-fucoidan preparation (550 mg twice a day for 48 weeks) was shown to have hepatoprotective effects related to serum concentrations of vitamin D, which is known to have immunoregulatory activity [109]. A clinical trial in healthy volunteers showed anti-inflammatory effects of a blend containing fucoidan from 3 different seaweeds. Daily oral ingestion of 1000 mg for 4 weeks was found to decrease serum IL-6 levels [110]. In advanced cancer patients a mixture of enzymatically digested and undigested fucoidan from *Cladosiphon*
*novae*
*caledoniae* (4 weeks of 4000 mg/day) was found to reduce several major proinflammatory cytokines, including IL-1β, IL-6 and TNF-α [111]. The analyses revealed that the responsiveness of IL-1β was inversely correlated with overall survival and was suggested as a possible prognostic factor for disease outcome in advanced cancer patients receiving fucoidan.

Taken together, brown seaweed polysaccharide ingestion is shown to be effective in antagonizing the effects of acute and chronic inflammation in both mouse models and clinical trials.

### 4.3. Fucoidan Ingestion and Atherosclerosis in Animal Models

Several studies have reported beneficial effects of fucoidan on outcome of atherosclerosis, a disease related to long-term inflammation of the arterial vessel wall.

ApoE-deficient mice are the most frequently used model for assessing atherosclerotic plaque development. In one study, sulphated polysaccharides from *Laminaria japonica* supplementation markedly reduced the thickness of the lipid-rich plaque, lipid peroxidation and foamy macrophage accumulation in the aorta via suppression of MAPKs and NF-κB signaling [112]. In line, Wang et al. found that *Laminaria japonica* fucoidan (50–100 mg/kg/day for 16 weeks) attenuated atherosclerosis by reducing inflammation and oxidative stress [113]. Furthermore, LMW fucoidan extracted from *Laminaria japonica* inhibited the formation of atherosclerotic plaques; and ameliorated the occurrence and development of atherosclerosis [114]. It decreased the production of inflammatory cytokines and prevented macrophages from developing into foam cells and diminished smooth muscle cells from migrating into the intimal layer of the aorta.

Furthermore, also in other models of atherosclerosis, fucoidan appears to have beneficial effects. In a rat allogenic aorta transplantation model ad libitum ingestion of fucoidan from *Fucus*
*vesiculosus* mediated anti-atherosclerotic activity by inhibiting inflammation, suppressing ROS production and down-regulating LOX-1 expression in the vascular wall [115]. In a rat aorta transplantation model fucoidan (LMW) from *Laminaria japonica* (200 mg/kg/day for 35 days) decreased the number of macrophages in the vascular wall by blocking P-selectin activity thereby preventing the development of aortic aneurysms [116]. In the LDLR^−/−^ mouse model of atherosclerosis *Laminaria japonica* fucoidan (50–100 mg/kg/day for 16 weeks) was shown to result in atherosclerosis attenuation by reducing inflammation and oxidative stress [113]. In conclusion, fucoidan appears to be promising in the battle against atherosclerosis by decreasing macrophage infiltration in the vascular wall, as well as by reducing inflammation and oxidative stress.

## 5. Phenolic Compounds and Phytosterols

### 5.1. Phytosterols

Phytosterols, including both sterols, stanols and oxysterols, such as fucosterol, saringosterol and 24-hydroperoxy-24-vinyl-cholesterol, are a group of functional lipid compounds. Compared to other bioactive molecules produced by brown algae, phytosterols exhibit various health-improving effects, especially neuroprotective and anti-inflammatory. Table A3 presents an overview of the significant inflammation-related outcomes in animal models after oral administration of phytosterols. Fucosterol, the most abundant sterol in brown seaweed, when administered in different animal models was found to induce a significant therapeutic effect on injury- or infection-related inflammation. Mo et al. [117] showed anti-inflammatory effects of fucosterol-pretreatment in Concanavalin A-treated mice as a model for acute liver injury. After Concanavalin A-treatment, NF-κB p65 increased markedly and the expression of a nuclear receptor in its upstream pathway, PPARγ, decreased. Both NF-κB p65 and PPARγ are closely related to the release of inflammatory factors such as TNF-α, IL-6 and IL-1β. Fucosterol pretreatment down-regulated the inflammatory response and subsequently necrosis and apoptosis by inhibiting the NF-κB pathway and activating PPARγ.

Anti-inflammatory effects of fucosterol were demonstrated using regular *Sargassum fusiforme* extracts (NH) and enzyme-modified *Sargassum fusiforme* extracts (EH) [22]. Enzyme modification significantly increased the fucosterol concentration in the extract (EH) leading to better results in decreasing pro-inflammatory cytokines as compared to the NH pretreatment group. In addition, both NH and EH reduced the production of NO without inducing any cytotoxicity and even increased cell viability in cultured RAW264.7 macrophages at a concentration of 10 μg/mL or higher. Anti-inflammatory effects of fucosterol were also observed in DNCB-induced NC/Nga mice as a model for atopic dermatitis. Oral administration of fucosterol significantly reduced [22] the scratching behavior of the mice and suppressed the production of pro-inflammatory cytokines (TNF-α and IL-4), resulting in reduced circulating IgE levels.

Bogie et al. showed that 24(S)-Saringosterol, an oxyphytosterol present in *Sargassum fusiforme*, has anti-inflammatory effects likely via activation of liver X receptor (LXR)β in a mouse model of Alzheimer’s disease (AD) [118]. *Sargassum fusiforme* extract rich in 24(S)-Saringosterol activated LXRβ preferentially and to a lesser extent also LXRα. LXRβ plays a key role in the down-regulation of the expression of multiple inflammatory genes [119,120]. AD is characterized by an accumulation of extracellular amyloid-β (Aβ), intracellular neurofibrillary tangles, loss of synapses, neuroinflammation and by a gradual progression of memory loss. After 45 days of dietary supplementation with *Sargassum fusiforme* the formation of Aβ plaques which is related to cognitive decline, was found to be dramatically reduced (~80% reduction) in AD mice. The expression of the LXR-target gene APOE in the central nervous system was increased due to administration of *Sargassum fusiforme* lipid extract. Apolipoprotein (Apo)E increased the clearance of Aβ by microglial cells and suppressed the secretion of Aβ by neurons in vitro. Therefore, the anti-inflammatory effects of 24(S)-Saringosterol may be explained by activation of the LXR-ApoE axis [121]. Similar to 24(S)-Saringosterol-mediated LXRβ activation, fucosterol can activate both LXRα and LXRβ, regulating different aspects of inflammatory gene expression.

### 5.2. Polyphenols

Polyphenols are another class of bioactive compounds from brown seaweed that have attracted great interest in recent years due to their pharmaceutical and biomedical properties. Polyphenols are classified based on their structure. Phlorotannins, highly abundant in brown seaweeds, are polymerized phenolic compounds consisting of phloroglucinol monomer units. Phlorotannins identified in brown seaweed, include eckol, dieckol, phlorofucofuroeckol A. Numerous studies have demonstrated the potential of polyphenol classes as antioxidant, anti-inflammatory, antidiabetic, antitumor, antihypertensive, anti-allergic, hepato-protective and anti-cancer. Table A4 provides an overview of the reported inflammation-related outcomes after oral administration of polyphenols in different animal models and in clinical trials. (Phase 0, 1, 2 and 3).

#### 5.2.1. Polyphenols: Effects in Steady State

The mechanisms underlying the anti-inflammatory effects of polyphenols are complex and are related to various stages of the inflammatory response that are sequential but overlapping. Disturbance of the steady state causes parenchymal tissue and immune cells to respond to injury or irritation through an innate cascade driving inflammation. Irfan et al. [122] demonstrated that phlorotannins strongly inhibit in vivo platelet aggregation in Sprague Dawley rats. In line with this in vitro, phlorotannins downregulated adenosine diphosphate-induced platelet activation (Ca-mobilization, fibrinogen binding, granule release—mediated via decreased Src, PI3K, PLCγ2, MAPK signaling). A clinical trial with 80 overweight participants showed that phlorotannins modestly decrease DNA damage [7], although no significant difference was found in acute phase proteins, anti-oxidant status or in inflammatory cytokines.

#### 5.2.2. Polyphenols: Effects on Allergy

NF-κB is one of the transcription factors that regulates eosinophilic inflammation and IgE-mediated hyperreactivity following allergic inflammation. Oral administration of polyphenols suppressed NF-κB pathway activation, and also inhibited inhibitor kappa B (IκB) that binds to NF-κB [123,124]. Polyphenols were found to alleviate particulate matter-induced airway inflammation in an allergic asthma mouse model [124]. Polyphenol treatment was found to decrease the inflammatory cell count in blood, including eosinophils, neutrophils, basophils. The level of epithelial cytokines, including IL-25, IL-33 and IL-8 also were reduced in the polyphenol-treated mice [124]. Han et al. [123] reported that pretreatment of BALB/c mice, as a model for passive cutaneous anaphylaxis, with Eckol inhibited the production of IL-4, IL-5, IL-6 and IL-13. Moreover, Eckol-treatment suppressed levels of β-hexosaminidase, secreted during the degranulation of mast cells. In addition, oral polyphenol administration was found to reduce the levels of FcεRI on the surface of IgE/bovine serum albumin (BSA)-stimulated mouse bone marrow-derived cultured mast cells (BMCMC). Cross-linking of FcεRI and allergen-specific IgE triggers allergic reactions that may be prevented by polyphenols. These results suggested that Eckol has anti-allergic potential. In a mouse ear-edema model both oral and local administration of phlorotannin, 1-21h prior to irritant application, strongly inhibited arachidonic acid (AA), 12-O-tetradecanoylphorbol-13-acetate(TPA)and immune-mediated (oxazolone (OXA))-induced ear swelling (30–80%) [125]. This suggests that the inhibitory effects of polyphenols are comparable to those of known anti-allergic agents. It was presumed that polyphenols play an anti-inflammatory effect by inhibiting mast cell degranulation, COX-2- and LOX-, and to lesser extent PLA2 activities.

#### 5.2.3. Polyphenols: Effects in Acute and Chronic Inflammation

Inflammation is a beneficial host response for foreign invaders and necrotic tissue with phase 1 being the first, immediate response, typified by ROS production and release of early mediators such as IL-1, TNF-α and arachidonic acid pro-inflammatory metabolites. Once detected extracellularly, ingested microbes will lead to upregulation of TLRs a family of proteins involved in the initial phase of host defense against invading pathogens. TLR4 is the most common member in inflammation phases. Excessive TLR activation, however, disrupts the immune homeostasis by sustained production of pro-inflammatory cytokines and chemokines.

Polyphenols were demonstrated to be very well capable of suppressing the increase of TLRs, including TLR4 [124,126,127], TLR2 [124,127] and TLR7 [124]. TLRs as primary sensors of microbial products activate signaling pathways that lead to the induction of immune- and inflammatory genes, such as the NF-κB pathway. Polyphenol treatment also significantly decreased NF-κB and thereby the modulation of inflammation-related signaling cascades [123,124,126,127,128,129,130,131].

ROS are a crucial factor in the inflammatory response, playing multiple roles after tissue injury, including initiation of acute inflammation, clarifying infection and necrotic tissue and mediation of various intracellular signal transduction pathways. Anti-inflammatory effects of polyphenols via antioxidant activities were demonstrated by Kang et al. [132]. They found that serum ferric reducing antioxidant power (FRAP) significantly increased 30 min after polyphenol treatment in the Sprague Dawley rat-model but declined quickly thereafter. Polyphenols showed anti-inflammatory effects by reducing the expression of ROS [131,133,134,135]. Administration of the polyphenol-rich fraction of *Ecklonia cava* reduced ROS and NO generation in LPS-stimulated inflammation in zebrafish [133]. ROS can activate a variety of transcription factors leading to the differential expression of genes involved in inflammatory pathways. On the other hand, excessive production of ROS can cause irreversible damage to DNA. Due to this dual effect, polyphenols often show a crucial effect in tumor models by upregulating ROS in tumor cells and at the same time downregulating ROS in healthy cells. Yang et al. [129] indicated that oral administration of phlorotannins to the SKOV3-bearing mouse model of ovarian cancer enhances cancer cell apoptosis via upregulation of the ROS pathway but protects against healthy kidney cell damage by downregulating ROS levels. Tissue damage leads to a rapid increase in ROS which stimulates PGE2 production via the activation of COX-2. Polyphenols exert anti-inflammatory effects not only by suppressing COX-2 [128,130,131,136], but also by reducing PGE2 production [130,131] which exacerbates inflammatory responses and immune diseases. The production of early inflammatory cytokines such as TNF-α, IL-6 and IL-1β, that is increased in phase 1, was significantly inhibited by oral administration of polyphenols in vivo, in different mouse, rat and zebrafish models [124,126,127,128,130,131,135,137,138,139], and also in vitro, in RAW264.7 macrophages [131,134]. Polyphenol treatment [128,138] decreased the expression of CCL2/MCP-1 and consequently may reduce the infiltration of macrophages and subsequently inflammation [126,128].

In phase 2, the regulation of inflammation is amplified via positive feedback and adaptive immunity is activated. Oral administration of polyphenols significantly affects macrophage infiltration and the balance of macrophages with an M1 or M2 phenotype. Oral administration of polyphenols decreased the expression of CD11b and CD80, markers for M1 macrophages [126,127,137,138]. The M2 type is identified by marker CD206, and can prominently expresses IL-10, a cytokine with potential anti-inflammatory effects and plays an important role in limiting the host immune response to pathogens. Polyphenol treatment was found to increase the expression of the M2 markers, CD206 [126,127,137,138] and IL-10 in acute liver injured mouse model and HFD with or without seaweed supplement mouse model [124,127,135,138]. Oral administration of polyphenols induced a decrease in the level of iNOS [130,131,134,138] and the levels of NO in both.

#### 5.2.4. Polyphenols: Late Consequences of Inflammation and Sequels

Oral administration of polyphenols may be promising for the treatment of severe or chronic inflammation and its consequences (phase 3). Oral administration of polyphenols resulted in a reduction in food intake and in body weight [126,127,128,137,140], as well as in the storage of triglyceride (TG) and total cholesterol (TC) [128,137]. After oral administration of extracts rich in polyphenols, also the leptin/adiponectin ratio, an important marker for inflammation and obesity, decreased [128]. Choi et al. and Son et al. [127,137] showed that the production of receptor for advanced glycation end-products (RAGE), closely related to inflammation and visceral fat hypertrophy, and RAGE-RAGE ligand binding was reduced in obese individuals treated with polyphenols. In obesity-associated type 2 diabetes, low-grade chronic inflammation can lead to an increase of blood glucose levels and to insulin resistance. Polyphenol treatment improved insulin sensitivity [128] and suppressed the increase of blood glucose levels in high-fat diet-induced obese mice [140]. A novel derivative from phloroglucinol called Compound 21 significantly exerted protective effects on multiple sclerosis through promotion of remyelination and suppressing neuroinflammation in a cuprizone-induced mouse model for multiple sclerosis [141]. In another study these authors showed that Compound 21 reduced the population of Th1/Th17 cells and inhibited their infiltration into the CNS. These results indicated a potential neuroprotective effect of Compound 21 [142].

In conclusion, polyphenols have various therapeutic effects including anti-inflammatory, anti-obesity, anti-diabetic and antioxidant. Polyphenols may be a highly promising treatment strategy for diseases involving chronic low-grade inflammation, but further clinical studies are needed.

## 6. Fucoxanthin(ol) and Meroterpenoids

Fucoxanthin is a major marine carotenoid present in chloroplasts of brown seaweeds, and particularly seaweeds such as *Undaria pinnatifida*, *Laminaria japonica* and *Sargassum honeri* are rich in fucoxanthin [143]. Ingested fucoxanthin is metabolized to fucoxanthinol in the small intestine and then absorbed [144]. Therefore, fucoxanthinol has a higher bioavailability than fucoxanthin. Multiple potentially beneficial health effects have been ascribed to both fucoxanthin and fucoxanthinol; e.g., anti-oxidative, anti-inflammatory, anti-obesity, anti-diabetic and anti-carcinogenic properties [145,146]. Meroterpenoids are partially derived from a terpenoid pathway (mero- means partial). Tetraterpenoids, of which carotenoids are the most common representatives, belong to the terpenoids and consist of eight isoprene units [147]. Meroterpenes can be isolated from brown algae such as *Sargassum serratifolium*. The meroterpenoid-rich fraction from the ethanolic extract (MES) of this brown alga is known for its antioxidant and anti-inflammatory activities [148]. Table A5 presents an overview of the significant outcomes of studies investigating oral administration of fucoxanthin, fucoxanthinol or MES in animal models related to the inflammatory response.

### 6.1. Fucoxanthin(ol) and Meroterpenoids: Effects in Acute and Chronic Inflammation

Fucoxanthin is known for its antioxidant potential through its ability to scavenge radicals effectively and to enhance enzymatic antioxidant activity [149]. Enhanced activity of antioxidant enzymes superoxide dismutase (SOD), catalase (Cat) and glutathione peroxidase (GPx) was observed in plasma and testis of rats [150], mice [151] and hamsters [152] after treatment with fucoxanthin. Fucoxanthin reduced the increased production of ROS as a consequence of increased oxidative stress and reduced the increased malondialdehyde formation during lipid peroxidation, which in turn causes upregulation of pro-inflammatory cytokine production [153,154]. Malondialdehyde levels were reduced in plasma, sperm and/or testicular tissue of rats [150], mice [151,155] and hamsters [152] after fucoxanthin treatment. Additionally, a reduction of ROS, such as superoxide (O_2_^−^) and hydrogen peroxide (H_2_O_2_) was seen both in vivo and in vitro after oral administration of fucoxanthin [150,151,152,155].

Oral administration of fucoxanthin in different animal models decreased the expression of various pro-inflammatory mediators, including cytokines such as TNF-α, IL-6 and IL-1β. This was observed in white adipose tissue, plasma, testis and colonic tissue after stimulation with various inflammatory triggers [150,151,155,156]. Sugiura et al. demonstrated the anti-inflammatory and inhibitory effects of oral or percutaneous administration of fucoxanthin on mouse ear swelling induced by different irritants [157]. Fucoxanthin and fucoxanthinol were shown to inhibit the enzymatic activities of PLA_2_ and COX-2, thus restraining the generation of pro-inflammatory arachidonic acid metabolites in these mice. These anti-inflammatory effects of fucoxanthin and fucoxanthinol were also confirmed in vitro using rat basophilic leukemia cells, which showed reduced mRNA expression of sPLA_2_ and COX-2 upon treatment with both fucoxanthin and fucoxanthinol [157]. In addition, Tan et al. found that COX-2 and iNOS mRNA expression were downregulated in obese mice upon fucoxanthin administration [155]. Similarly, Maeda et al. demonstrated decreased expression of MCP-1 in white adipose tissue in mice with obesity-related inflammation upon treatment [145,156]. Since MCP-1 is a pro-inflammatory cytokine this suggests an anti-inflammatory effect of fucoxanthin on adipocytes. In mouse models of DSS-induced colitis and colitis-associated colon carcinoma, a reduction in total NO content and in NO release in colonic tissue was observed after oral administration of fucoxanthin [151]. Moreover, NO production was also reduced after treatment with fucoxanthin in cisplatin-induced testicular damage in hamsters [152]. Additionally, the meroterpenoid-rich fraction of an ethanolic extract from *Sargassum serratifolium* (MES) induced anti-inflammatory activities in high-cholesterol-fed mice. The mice demonstrated decreased serum levels of MCP-1 and keratinocyte chemoattractant, which are pro-inflammatory chemokines causing monocyte adhesion in vascular lesions. Furthermore, MES supplementation resulted in reduction of ICAM-1, VCAM-1, MCP-1, COX-2 and MMP-9 expression in aortic tissue. These results indicated that MES prevented vascular inflammation in these mice [148]. Similarly, mice fed a high-fat diet supplemented with MES, compared to un-supplemented high-fat diet showed decreased expression of macrophage markers F4/80 and MCP-1, indicating a suppression of inflammation [158].

### 6.2. Fucoxanthin(ol) and Meroterpenoids: Late Consequences of Inflammation and Sequels

Related to the inflammation in adipose and other tissues, oral fucoxanthin supplementation was also shown to have effects counteracting obesity and obesity-related morbidity [159]. Administration of fucoxanthin to mice fed a high-fat diet reduced gain in body weight and in weight of white adipose tissue as compared to chow-fed control mice [155,156]. Furthermore, Maeda et al. demonstrated that mice fed a high-fat diet also receiving fucoxanthin displayed significantly lower plasma levels of LDL-cholesterol and leptin compared to mice that were fed a high-fat diet only, indicative of moderated metabolic dysregulation [156]. Additionally, the obesity-related reduction in expression of beta-3 adrenergic receptor (ADRB3), responsible for lipolysis and thermogenesis [160], appeared significantly restored in mice upon addition of fucoxanthin to their high fat diet [156]. Moreover, Tan et al. showed a decrease in myeloperoxidase (MPO) activity in mice with high-fat diet-induced obesity after oral fucoxanthin administration, which implies a reduction in polymorphonuclear cell infiltration [155]. In addition, MES supplementation suppressed body weight, TG, glucose and free fatty acid concentrations in plasma of high fat diet-fed mice. In addition, the lower HDL cholesterol levels increased to comparable levels as in the control group. Moreover, increased expression of UCP-1 and ADRB3 in subcutaneous tissues demonstrates that MES supplementation causes conversion of white to beige/brite adipocytes which resembles brown adipose tissue [158]. These results suggest anti-obesity effects and inhibition of lipogenesis by MES supplementation.

In line with improvement of metabolic functions induced by fucoxanthin supplementation, anti-diabetic effects of fucoxanthin have been observed [145,150,156]. Feeding a high fat-diet containing fucoxanthin resulted in decreased plasma insulin and blood glucose levels in mice, to levels similar as in mice fed a regular diet [156]. Moreover, mRNA levels of GLUT4, encoding the insulin-sensitive glucose transporter in adipose tissue and muscle, were restored to normal levels when the high fat diet was supplemented with fucoxanthin [156]. In the diabetic KK-Ay mouse model fucoxanthin consumption was found to decrease elevated plasma blood glucose concentrations [145]. It was also shown that glucose intolerance improved by fucoxanthin dietary addition [145]. More recently, Kong et al. found that treatment of diabetic rats with fucoxanthin significantly reduced levels of plasma glucose compared to diabetic rats without any treatment [150]. Insulin concentrations and homeostatic model assessment of insulin resistance (HOMA-IR) levels were significantly reduced in these rats. Finally, fucoxanthin supplementation inhibited the expression of the suppressor of cytokine singaling-3 (SOCS-3), involved in the induction of insulin resistance [150]. Together, these results indicate that fucoxanthin possesses anti-diabetic effects by suppressing inflammation and thereby improving insulin sensitivity.

## 7. Concluding Remarks

In this comprehensive systematic review, we aimed to provide an overview on the modulating role of intake of complete brown seaweed, its extracts or selected compounds on the modulation on different aspects of the inflammatory immune response. This includes the impacts of seaweed consumption on steady state immune parameters, effects on allergies, the innate and adaptive immune response and also on chronic, low-grade inflammation. Brown seaweeds constitute a group of approximately 2000 species containing several common but also unique bioactive molecules with immunomodulatory functions. We therefore distinguished the impact of four different categories of compounds, i.e., polysaccharides, (poly)phenols, phytosterols and carotenoids.

We identified three common denominators across the effects of brown seaweed and constituents thereof on inflammation (represented in Figure 2). Firstly, most purified compounds, despite their diverse chemical nature, appear to inhibit similar aspects of inflammation, in particular synthesis of reactive oxygen species. This common effect is rather puzzling. Yet, a caveat of some studies may be the use of compound concentrations beyond physiological levels. In addition, purification of the compounds used in the cited studies mostly left sufficient room for concentrations of contaminants that might contribute to, or even explain, the claimed effects. In our view, this calls for rigorous comparative research of the various compounds in an identical experimental setting. Nevertheless, the suppression of reactive oxygen species provides an interesting angle to study immune-modulatory effects of brown seaweed constituents.

Secondly, brown seaweeds interfere with the innate immune response on the level of TLR-induced NF-κB signaling. This route is actually linked with the previous one since oxygen radicals drive and amplify innate NF-κB-mediated activation. In conjunction, a plethora of in vitro and in vivo experiments support a suppression of IL-6, IL-1, iNOS and TNF-α upon treatment with brown seaweeds. Polyphenols, fucosterol, fucoxanthin and fucoidan all seem to be active in this. Yet, when fucoidan was applied in the pretreatment setting, it was also shown to potentiate the NF-κB axis to reduce susceptibility to infection via scavenger receptor A and TLR4 activation in an antibiotic-like fashion. These seemingly contradictory findings underscore the versatile properties of brown seaweed constituents in the modulation of the innate immune response. Therefore, interpretation should be performed with care when extrapolating in vitro findings to human applications. Nevertheless, most in vivo studies summarized in this review show unequivocal anti-inflammatory effects. The mechanistic background of brown seaweed health benefits probably goes beyond direct inhibition of inflammation. Multiple studies indicate that brown seaweed compounds interact with pathways and processes involved with energy sensing and survival, such as AKT/mTOR/AMPK and autophagy. Thus, brown seaweed compounds might also function as caloric restriction mimetics and thereby stimulate vitality.

Thirdly, the adaptive immune response displays an altered Th1/Th2 response in response to brown seaweed. Different constituents have been identified to skew the Th1/Th2 balance. Depending on the molecular weight of the fucoidan, different outcomes have been identified. Fucosterol has been shown to skew Th0 cells into Th2 cells in a model for allergy, whereas polyphenols suppress the Th1/Th17 response in an animal model for MS. The net effect of brown seaweed on Th1/Th2 skewing cannot be generalized and is largely dependent on the composition of the different constituents in the seaweed. On the level of allergy, Th2 suppression reduces IL-4 cytokine levels, decreased IgE production and suppressed mast cell activity.

Inflammation is a universal response of the body to damage, infection or otherwise disturbed homeostasis. For this review, we have restricted the search terms to those that are related to inflammation, allergy and immunity. The broad implication of the inflammatory response in the maintenance of bodily integrity, however, entails that several studies that focus on aspects only indirectly related to inflammation, were not included in the final result, whereas similar studies were, based on different choices by the respective authors for their specific wording. Nevertheless, we are confident that this review covers the main aspects of oral supplementation of brown seaweeds and their components on aspects of inflammation, allergy and immunity in a broad sense. More well-designed human studies applying individual seaweed constituents as well as whole seaweed (extracts) will provide more insight into the applicability of brown seaweed as immune-modulatory nutritional intervention strategies.

## Figures and Tables

**Figure 1 nutrients-13-02613-f001:**
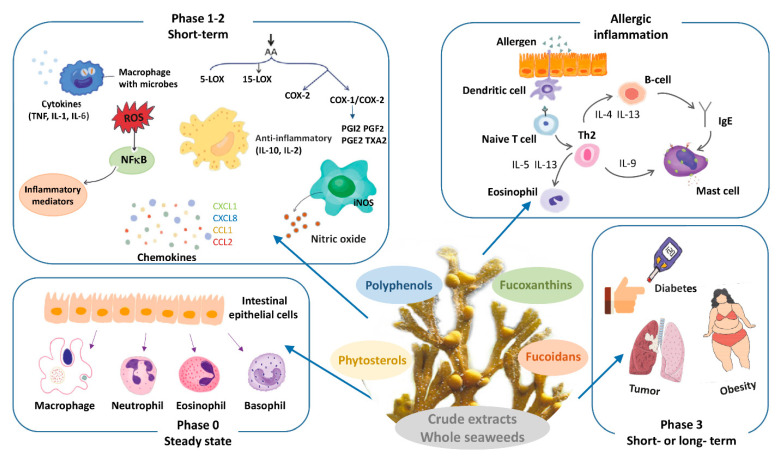
Schematic representation of brown seaweed and it constituents and of the phases in the immune response (Phase 0: steady state, allergy, Phase 1–2: acute inflammatory response and Phase 3: chronic inflammatory response) addressed in this review.

**Figure 2 nutrients-13-02613-f002:**
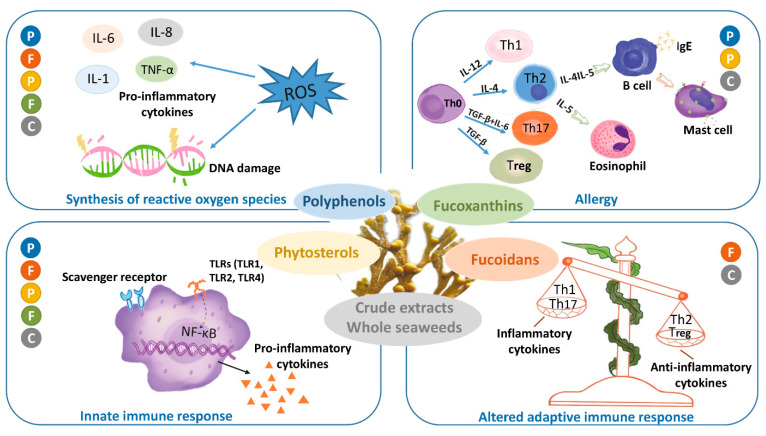
Overview of the general effects of crude brown seaweed or its extracts (C, grey) and of its constituents: polyphenol (P, blue), Fucoxanthin (F, green), Fucoidan (F, brown) and Phytosterols (P, yellow) on the different phases of the inflammatory processes, including allergy. The colored dots showing the crude seaweed or its extract and its constituents in the separate boxes indicate in which phases of the immune response they exert their effects.

**Table 1 nutrients-13-02613-t001:** Phases of the inflammatory response with characteristic features ^(1)^.

Phase	Description	Characteristic Read-Outs
0 steady state	homeostatic condition	- growth, organ weight, etc. - leukocyte numbers and subset composition - steady state activities, e.g., phagocytosis
1 short-term	initiation of inflammation as response to triggering by damage- or infection-related molecules	- mast cell and basophil degranulation - neuronal activation (e.g., scratching) - reactive oxygen species (ROS) - pro-inflammatory arachidonic acid (AA) metabolites (e.g., PGE_2_) - phospholipase (PLA_2_), cyclo-oxygenase (COX-2/PTGS2) and lipoxygenase (LOX/ALOX) activity - early pro-inflammatory cytokines (IL-1, TNF, IL-6 ^(2)^)
2 short-term	amplification and regulation of inflammation; initiation of adaptive immunity	- additional pro- and anti-inflammatory cytokines (e.g., IL-12, IL-10, IL-2) - soluble forms of cellular R (e.g., sCD25, sCD163) - chemokines (e.g., CXCL8, CCL2) - induction of iNOS and nitric oxide (NO) production - anti-inflammatory AA metabolites (e.g., LXA4) - endothelial activation - edema - leukocyte mobilization and tissue infiltration - acute phase proteins (e.g., CRP) - HPA-axis activation (cortisol or corticosterone) - lymphocyte proliferation - microbial infection parameters - clearance and repair in short-term
3 short- or long-term	consequences of severe or chronic inflammation	- clearance and repair in long-term - glucose, insulin resistance, diabetes - adipokines, leptin resistance, obesity - glucocorticoid resistance, stress, hampered down-regulation of inflammation - inflammation-induced malignancy

^(1)^ See Box 1 for a brief introduction on inflammation. ^(2)^ In the inflammatory response, IL-6 is produced in a second wave, as it has to be expressed de novo, while initial TNF and IL-1β release only requires processing (i.e., membrane cut or enzymatic cleavage and secretion, resp.). Yet, IL-6 is frequently measured together with IL-1β and TNF. Therefore, IL-6 is classified in the first phase. In itself, it is an inducer of mediators of the second phase, in particular acute phase proteins.

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
