# Peer review of "Brown Seaweed Food Supplementation: Effects on Allergy and Inflammation and Its Consequences"

_nutrients, 2021, doi:10.3390/nu13082613_

Round 1

Reviewer 1 Report

I would like to thankthe authors for this work. It's a well documented review, need to be in some places more synthetic but the review is well written and very rich. This work will be benefic to the scientific community.

I have some comments and questions to the authors:

  • Please homogenize the terme laminaran and laminarin in the text
  • No information concerning the detailed structures of the polysaccharides such as fucoidans?
  • The bioactivity related to MW of fucoidans was reported, I was wonderring if it was also reported any biological effect depending on the degree of polymerization of polysccharides such as fucoidans?
  • Please correct the title of the figure 2 in the end of the sentence replace "the" by "they exert"

Author Response

  • Please homogenize the terme laminaran and laminarin in the text. Laminaran has been changed to laminarin now, as suggested.
  • No information concerning the detailed structures of the polysaccharides such as fucoidans? We have added a sentence with more information: "Fucoidans are a group of polysaccharides (fucans) primarily composed of sulphated L-fucose with less than 10 % of other monosaccharides. They are widely found in the cell walls of brown seaweed, but not in other algae or higher plants (Berteau and Mulloy 2003)".
  • The bioactivity related to MW of fucoidans was reported, I was wonderring if it was also reported any biological effect depending on the degree of polymerization of polysccharides such as fucoidans? For this review we consider discussing the effects of the degree of polymerization too complex. Fucoidans have been hydrolyzed using different enzymes or acid and were subsequently tested for MW and not for degree of polymerization. 
  • Please correct the title of the figure 2 in the end of the sentence replace "the" by "they exert" this has been adapted in the revised version of the manuscript.

Reviewer 2 Report

The manuscript presents an interesting study on the use of brown seaweeds as food additives. It is a comprehensive and well structured review.

Seaweeds represent a vast resource that remains underutilized as an ingredient, and has become a billion dollar industry and has reached the forefront of talks regarding food sustainability and supply chains.

There is a growing interest that bioactive compounds from seaweed can play a major therapeutic role in disease prevention in humans. The present manuscript critically examines the existing scientific knowledge on primary and secondary metabolites from seaweed and their functional properties for some health-related conditions, and their therapeutic development is discussed.

Author Response

We would like to thank the reviewer for the positive comments.